# CReST: Cross-Query Residual Transformer for Object Goal Navigation

## Abstract

Object Goal Navigation (OGN) is the task of navigating from a random location to the target objects in an unknown environment. The end-to-end navigation method decides the actions of the agent to navigate to the target objects and relies much on the state representation obtained by the visual information processing network. In this paper, we propose the Cross-Query Residual Transformer (CReST) to extract more sufficient visual features from the image of the RGB camera for better navigation results, which includes the overall network, Residual Transformer, and Random Mask. In the overall network, the Global Feature and the Local Feature mutually query each other and are subsequently fused for better visual information processing. The Residual Transformer adds residual connections to the Transformer to solve the gradient vanishing problem, which enables the whole network to be trained in one stage without pretraining and allows the Transformer to be several times deeper. The Random Mask is proposed for data augmentation and overfitting reduction. The experiments demonstrate that CReST surpasses the competing methods and achieves state-of-the-art performance on the AI2-THOR dataset. The ablation experiments prove the Residual Transformer and the Random Mask contribute much to the navigation results.

## 1 Introduction

Object Goal Navigation (OGN) is a navigating task that starts from a random location to the objects of the target category at unknown locations in an unknown environment. The key to this problem is searching target objects with local and limited information, which is very different from the conventional navigation methods that are based on the known locations of target objects and a prior map of the environment.

The end-to-end navigation method for object goal navigation decides the action from the visual information without any map or path planning, which is similar to the researches that show deep learning can exceed humans without conventional human knowledge Silver et al. (2017b) Silver et al. (2017a). The end-to-end navigation method encodes the visual information into the state representation and uses it to decide the actions. The navigation performance relies on the useful information in the state representation, which means the visual information encoding determines the navigation result.

SAVN Wortsman et al. (2019) encodes the visual information with a ResNet18 He et al. (2016) pretrained on ImageNet Russakovsky et al. (2015), and Spatial Attention Mayo et al. (2021) encodes both semantic information about the observed objects and spatial information about their place using a convolutional network. ORG Du et al. (2020a) improves visual representation learning by integrating object relationships, including category closeness and spatial correlations. VTNet Du et al. (2020b) uses supervised learning to extract the visual feature to the state representation and adopts reinforcement learning to decide the action.

In this paper, we propose the Cross-Query Residual Transformer (CReST) to extract the visual features from the image of the first perspective RGB camera for better navigation results, which includes the overall network, the Residual Transformer, and the Random Mask. In the overall network, the Global Feature and the Local Feature mutually query each other and are subsequently fused for better visual information processing. The Residual Transformer adds residual connections to the

Transformer to solve the gradient vanishing problem. The Random Mask is used for data augmentation and overfitting reduction. The experiments demonstrate that CResT surpasses the competing methods and achieves state-of-the-art performance on the AI2-THOR Kolve et al. (2022) dataset. The contributions of this paper are summarized as follows.

- In this paper, we propose CResT to process the visual information for improving the results of the object goal navigation, in which the Global Feature and the Local Feature query each other and are subsequently fused to the State Representation. In this way, the CResT can extract more useful visual information from the image of the RGB camera.

- The Residual Transformer is proposed to solve the gradient vanishing problem. The Residual Transformer adds residual connections to the standard Transformer for better backpropagation, which enables the whole network to be trained in one stage without pretraining and achieve better performance.

- We propose the Random Mask for data augmentation in the network that can reduce overfitting. This module efficiently augments the Global Feature and the Local Feature and enhances the generalization performance for better navigation results.

## 2 RELATED WORKS

### 2.1 OBJECT GOAL NAVIGATION

In the task of object goal navigation, the agent needs to search for a target object described by text, image, or label in the environment Li et al. (2023). The object goal navigation solutions include the end-to-end method and the map-based navigation method. The end-to-end navigation method decides the action from the visual information without any map, which is capable and concise. On the other hand, the map-based navigation method builds the map Klein & Murray (2007), Mur-Artal & Tardós (2017), Engel et al. (2017), Liu et al. (2016),Chaplot et al. (2019), estimates a long-term goal Chaplot et al. (2020a), Ramakrishnan et al. (2022), Zhu et al. (2022), Chaplot et al. (2020b), Campari et al. (2022), Wu et al. (2022), Georgakis et al. (2022), Chaplot et al. (2021), Staroverov et al. (2020), and plans a path from the current location to the long-term goal Wang et al. (2011), Hart et al. (1968), Sethian (1996).

### 2.2 END-TO-END METHOD

The end-to-end navigation method encodes the visual information to the state representation, which is used to decide the actions. In this method, the visual information processing network is significant to the navigation results. The Transformer Vaswani et al. (2017) is used to process images and achieves high accuracy, such as ViT Dosovitskiy et al. (2020) and DETR Carion et al. (2020), and it can also be applied to the object goal navigation because of its powerful self-attention ability.

The end-to-end navigation method research has experienced a fast development in recent years. SAVN Wortsman et al. (2019) proposes a self-adaptive visual navigation method (SAVN) that learns to adapt to new environments without any explicit supervision. Spatial Attention Mayo et al. (2021) proposes an attention probability model to encode the semantic and spatial information of observed objects and enables the agent to navigate to target objects efficiently after the training of reinforcement learning. Du et al. Du et al. (2020a) proposed three complementary techniques to enhance the robustness and enable the agent to escape from deadlock, which are object relation graph (ORG), trial-driven imitation learning (IL) and a memory-augmented tentative policy network (TPN).

Some end-to-end navigation methods employ the graph convolutional network. HOZ Zhang et al. (2021) builds a graph called Hierarchical Object-to-Zone (HOZ) to represent rooms from overall to objects. Pal et al. Pal et al. (2021) represents the proximity between the agent and target object by graph neural network and makes full use of the prior information of the scene to guide the agent to the target. Lyu et al. Lyu et al. (2022) uses a graph convolutional network to encode spatial relationships and deep reinforcement learning to make decisions.

The Transformer effectively improves the navigation result due to its mighty self-attention. VTNet Du et al. (2020b) decides action from the visual information in two stages: Visual Transformer and Navigation Strategy Network. The Visual Transformer processes the visual information to the state

representation with supervised learning, and the navigation strategy network decides action from the state representation with reinforcement learning. OMT Fukushima et al. (2022) expands the visual information to a period and improves the network to process multi-time data with the Transformer, so the network has 'time memory' for navigation.

# 3 CROSS-QUERY RESIDUAL TRANSFORMER (CREST)

In this paper, we propose the Cross-Query Residual Transformer (CResT) to encode the visual information for better navigation results, which includes the overall network, the Residual Transformer, and the Random Mask.

## 3.1 OBJECT GOAL NAVIGATION SCENARIO

In the end-to-end method for object goal navigation, the agent decides its action from the visual observation and the target category at every step Li et al. (2023), which includes the visual encoding network and the navigation policy. The visual information is encoded into a feature map, called state representation, which can be used as the input of the navigation policy network that decides the final action.

The agent can reach one of the target objects step by step without the map of the environment or the locations of the target category if only its action always approaches the location of the target objects. The only sensor of the agent is an RGB camera, and its action space $A$ consists of 6 actions, i.e., $A = \{MoveAhead, RotateLeft, RotateRight, LookUp, LookDown, Done\}$. At every time step, the agent can move forward for 0.25 meters, turn left or right for 45 degrees, or look up or down for 30 degrees. The searching episode is judged as success when the agent reaches the area within 1.5 meters of the target object, sees the target object from its camera, and takes the action of 'Done'. Otherwise, the episode is judged as a failure. When there are several objects of the target category in the environment, the agent only needs to search for one of them to finish the task. In this kind of method, the visual encoding network is critical to the navigation results, because it extracts the visual features from the RGB sensor and decides the input of the navigation policy network.

## 3.2 OVERALL

In this paper, we propose CResT for visual encoding in the end-to-end navigation method. The CResT processes the visual information to the State Representation, as shown in Figure 1. The Global Feature and Local Feature are extracted from the image of the RGB camera and are input to Residual Transformer (Global) and Residual Transformer (Local) for cross-querying each other, after which the outputs of both Residual Transformers are fused to the State Representation by Residual Transformer (Embedding). Then the State Representation can be used as input for the Navigation Policy Network for action decisions.

The Global Feature is extracted from the whole image by Convolutional Neural Network (CNN), and the Local Feature Du et al. (2020b) is extracted from the objects detected by a trained DETR Carion et al. (2020). The Global Feature and Local Feature are partially masked by the Random Mask module for data augmentation, which can reduce overfitting in the deep neural network.

The Global Feature and the Local Feature are input to the encoder and the decoder respectively in Residual Transformer (Global) for cross query, as shown in Figure 2. The encoder of the Residual Transformer encodes its input, i.e., the Global Feature, with self-attention, and the decoder processes the output of the encoder and another input of the decoder, i.e., the Local Feature, with cross-attention, so the Global Feature is queried by Local Feature in Residual Transformer (Global). In the same way, the Local Feature and the Global Feature are input to the encoder and the decoder respectively in Residual Transformer (Local) for cross query, so the Global Feature and the Local Feature can query each other in both Residual Transformers. After the mutual queries, the outputs of both Residual Transformers are jointly embedded into the State Representation by the Residual Transformer (Embedding).

The overall network can be trained efficiently by the reinforcement learning in one stage, which is more concise and efficient than the two-stage training in VTNet Du et al. (2020b). Then the

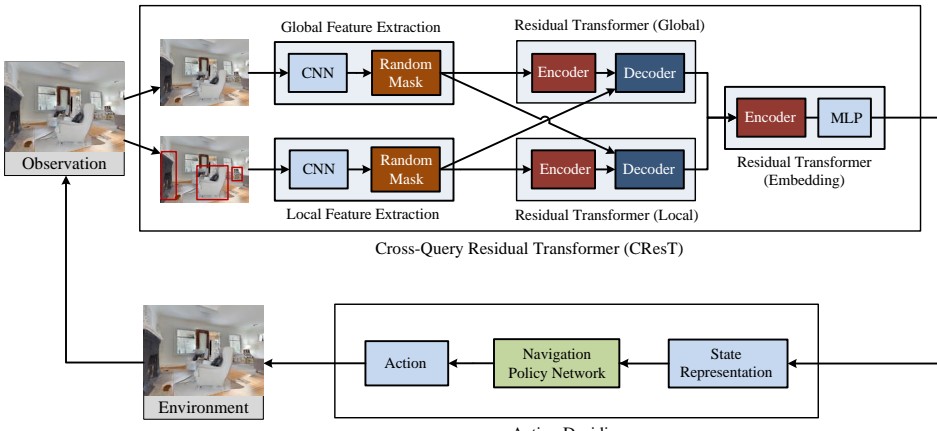

Figure 1: The Cross-Query Residual Transformer (CResT). The Global Feature and Local Feature query each other and are fused for better visual feature extraction. The Residual Transformer efficiently solves the gradient vanishing problem, and the Random Mask improves visual features in the way of data augmentation.

agent can decide the navigation action from the visual information directly without the map of the environment or the locations of the target objects.

## 3.3 RESIDUAL TRANSFORMER

The Transformer obtains excellent performance in Natural Language Processing (NLP) and Computer Vision (CV). The VTNet Du et al. (2020b) must be trained in two stages and becomes totally not trainable when the number $N$ of the Transformer layers increases from 6 to 8. Fortunately, the network recovers to trainable again when $N$ decreases or the residual connections are added, so the problem in VTNet can be recognized as the gradient vanishing problem.

The gradient vanishes in deep neural networks because of the chain rule in gradient computing, which makes the neural network almost unchanged in training. This problem is solved by residual connection in ResNet He et al. (2016), and then the neural network can be increased to over one thousand layers. The residual connection is widely used in deep learning networks such as Transformer, as shown in Figure 2. However, the residual connection in the standard Transformer is not enough, such as the lack of residual connection between the output of the encoder and the output of the decoder, which may cause the gradient vanishing problem.

The Residual Transformer proposed in this paper adds 3 residual connections to the Transformer, in which Residual connection 1 (C1) connects the input with the output in the encoder, Residual connection 2 (C2) connects the output of the encoder with the output of the decoder, and Residual connection 3 (C3) connects the inputs2 with the output in the decoder, as shown in Figure 2. These residual connections substantially enhance the backpropagation in the Transformer and effectively solve the gradient vanishing problem. The Residual Transformer enables CResT to be trained in one stage instead of the two-stage training in VTNet Du et al. (2020b) because the residual connection in the Residual Transformer can efficiently facilitate the backpropagation in the network. This one-stage training is more concise and can achieve better performance than the two-stage training.

The dashed line in the Residual Transformer means the size of the feature map changed through the residual connection, as shown in Figure 2. The sizes of feature maps are different between the output of the encoder and the output of the decoder, so they cannot be added directly through the residual connection. This may be the reason why the residual connection is not used to connect the output of the encoder and the output of the multi-head attention of the decoder in the standard Transformer. However, our experiments show the size of the feature map can be adjusted by a

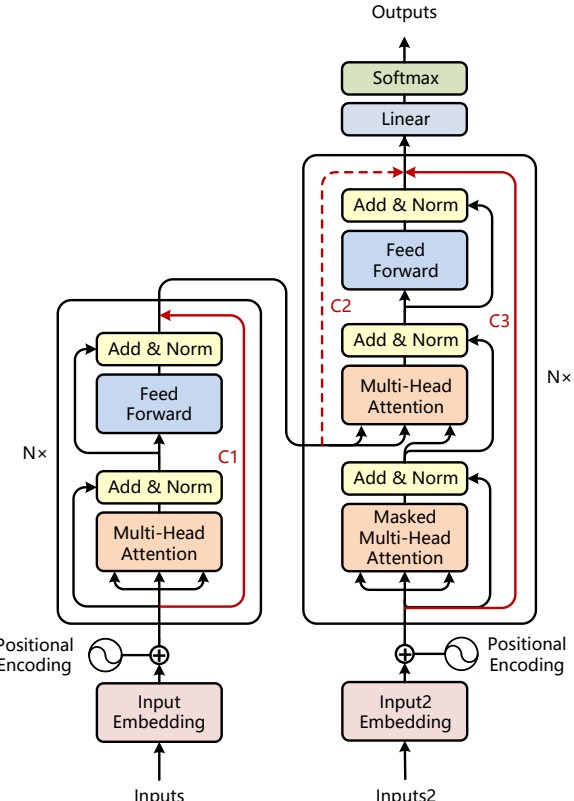

Figure 2: Residual Transformer. The Residual Transformer can efficiently solve the gradient vanishing problem, in which three residual connections are added to the standard Transformer. Residual connection 1 (C1) connects the input with the output in the encoder, Residual connection 2 (C2) connects the output of the encoder with the output of the decoder, and Residual connection 3 (C3) connects inputs2 with the output in the decoder.

convolutional neural network without losing the feature information. The channel number $c1$ of the feature map can be changed to $c2$ by the $1 \times 1 \times c1 \times c2$ convolution kernel without changing the feature information, which can be used to adjust any feature map dimension by transposing it before computing. For example, the size of the feature map can be adjusted from $m \times n1 \times c$ to $m \times n2 \times c$ by the $1 \times 1 \times n1 \times n2$ convolutional kernel. Firstly, we can transpose the feature map between dimension 2 and dimension 3, which changes the size from $m \times n1 \times c$ to $m \times c \times n1$. Then the transposed feature map computes convolution with $1 \times 1 \times n1 \times n2$ convolutional kernel, which turns the size from $m \times c \times n1$ to $m \times c \times n2$. Finally, the feature map is transposed back to $m \times n1 \times c1$ and finishes the size transformation. In this way, the feature map can be adjusted to any size using a convolutional kernel without changing the feature information, which enables the residual connection to add feature maps of different sizes.

## 3.4 RANDOM MASK

The Random Mask module is proposed in this paper to solve the overfitting problem to obtain better results. Overfitting means wonderful training results yet unsatisfying testing results, and it becomes obvious when a relatively larger network trains on a relatively limited dataset. Data augmentation can solve this problem, such as rotating or masking images in computer vision. The Random Mask module can mask the feature map randomly to make it different from before. Because the masked part is always random, the masked feature map is always different at each time, which augments

the dataset better than the fixed mask. The Random Mask modules are used to augment the Global Feature and Local Feature, as shown in Figure 1.

Firstly, the Random Mask module generates a uniformly distributed random array $M = \{m|m \in [0,1]\}$ that is the same size as the input feature map, and the threshold is set to differentiate the elements in the array. Then the element $m$ is set to 1 if it is larger than or equal to the threshold, or it is set to 0 otherwise. Then the random array $M$ turns to a binary array $B = \{b|b = 0,1\}$.

$$b = \begin{cases} 1 & m \geq \text{threshold}, \\ 0 & \text{otherwise}. \end{cases} \tag{1}$$

Finally, the input feature map is multiplied by the corresponding elements of the binary array. The elements in the feature map corresponding to 1 in the binary array remain invariable, while the elements corresponding to 0 become 0, which randomly masks the input feature map. The proportion of the masked part can be adjusted by setting the threshold. For example, approximately 10% of the feature map is masked if threshold = 0.1. This Random Mask module can efficiently augment the Global Feature and the Local Feature and greatly improve the generalization performance of the network, which is verified in the following experiments.

## 4 EXPERIMENTS

### 4.1 EXPERIMENTAL CONDITIONS

The hardware conditions of the experimental platform are Intel (R) Xeon (R) Gold 5117 CPU with four NVIDIA GeForce RTX 2080 Ti GPU, and the software conditions are Ubuntu 22.04, Docker 20.10.21, Python 3.6.13, and PyTorch 1.4.0. The dataset AI2-THOR Kolve et al. (2022) is a 3D simulation environment with realistic photos, which contains four types of scenes: living room, kitchen, bathroom, and bedroom. There are 30 different rooms with various furniture and other objects in each type of scene, a total of 120 rooms. In these rooms, 80, 20, and 20 rooms are used as the train set, validation set, and test set respectively.

CReST is trained in one stage instead of the two-stage training in VTNet Du et al. (2020b) because the Residual Transformer facilitates the backpropagation. In the training, the penalization of each action step is -0.001, and the reward of the successful episode is 5. This reward strategy leads the agent to pursue success in the shortest path. In the experiments of CReST and competing methods, the default mask threshold is 0.1, the default layer number of the Transformer is 6, and the number of training episodes is $2 \times 10^6$.

### 4.2 EVALUATION METRICS

The metrics in the experiments are Success Rate (SR), Success weighted by Path Length (SPL), and Distance To Success (DTS). Success Rate (SR) is the percentage of successful episodes in all exploration episodes:

$$SR = \frac{1}{N} \sum_{i=1}^{N} S_i \tag{2}$$

Where $S_i$ indicates whether the i-th episode is successful, and $N$ is the total number of episodes. Success weighted by Path Length (SPL) measures the efficiency of the exploration:

$$\text{SPL} = \frac{1}{N} \sum_{i=1}^{N} S_i \frac{l_i}{\max(p_i, l_i)} \tag{3}$$

Where $S_i$ indicates whether the i-th episode is successful, and $N$ is the total number of episodes. $l_i$ is the length of the shortest path from the initial position to the target objects, and $p_i$ represents the length of the path traveled by the agent in the i-th episode. This means the closer SPL is to 1, the more efficient the exploration is.

Table 1: Experimental results. The table shows the Success Rate and the SPL of CResT and the competing methods in all episodes and the episodes of step $L \geq 5$ that show the performance of the relative far searching task.

| Method | All | | $L \geq 5$ | |
|---|---|---|---|---|
| | Success | SPL | Success | SPL |
| Spatial Attention | 0.4050 | 0.1383 | 0.2964 | 0.1307 |
| SAVN | 0.4350 | 0.1610 | 0.2919 | 0.1366 |
| ORG | 0.6684 | 0.3432 | 0.5700 | 0.3265 |
| ORG+TPN | 0.6708 | 0.3839 | 0.5661 | 0.3639 |
| VTNet | 0.6976 | **0.4194** | 0.5831 | 0.3892 |
| CResT | **0.7322** | 0.4061 | **0.6420** | **0.3935** |

## 4.3 COMPETING METHODS

We compare our method with the following methods: Self-adaptive Visual Navigation (SAVN) Wortsman et al. (2019), Spatial Attention Mayo et al. (2021), ORG Du et al. (2020a), ORG+TPN Du et al. (2020a), and VTNets Du et al. (2020b).

**SAVN** Wortsman et al. (2019) proposes a self-adaptive visual navigation method (SAVN) that learns to adapt to new environments easily in testing. This meta-reinforcement learning approach enables the agent to learn a self-supervised interaction loss that remains fixed while the rest of the network can be updated in testing, which helps the network adapt to the testing environment without extra supervision.

**Spatial Attention** Mayo et al. (2021) uses an attention probability model in the agent's policy trained with reinforcement learning. The attention model consists of three components: target, action, and memory, which encodes semantic information about observed objects and spatial information about their place. This mechanism helps the agent to navigate to the target object effectively.

**ORG+TPN** Du et al. (2020a) proposes three complementary techniques: object relation graph (ORG), trial-driven imitation learning (IL), and a memory-augmented tentative policy network (TPN). ORG improves visual encoding by using object relationships, such as category closeness and spatial correlations. Meanwhile, IL and TPN help the agent to escape from deadlock states, such as looping and being stuck.

**VTNet** Du et al. (2020b) proposes a Visual Transformer Network (VTNet) for learning informative visual representation in navigation, which encodes the global feature and the local feature from RGB images. Meanwhile, VTNet embeds object and region features with their locations as spatial-aware descriptors to informative representation, and then agents can explore the correlations between visual observations and navigation actions.

## 4.4 EXPERIMENTAL RESULTS

The navigation results of CResT and competing methods are shown in Table 1. The navigation performance of CResT exceeds that of competing methods and thus achieves state-of-the-art performance on the AI2-THOR data.

**Comparing to Spatial Attention Mayo et al. (2021) baseline.** Spatial Attention uses both the semantic information and the spatial information of observed objects in visual encoding, but it gains an unsatisfying result because of its lightweight convolutional neural networks for visual encoding. This type of visual encoder makes them inferior to other competing methods. CResT greatly sur-

Table 2: Ablation experiments. The experiments show the efficiency of the components in CResT, and all the methods are trained in one stage.

| Methods | All | | $L \geq 5$ | |
|---|---|---|---|---|
| | Success | SPL | Success | SPL |
| Layer 1 | 0.7312 | **0.4073** | 0.6254 | 0.3930 |
| Layer 6 | **0.7322** | 0.4061 | **0.6420** | **0.3935** |
| VTNet (1 stage) | 0.0845 | 0.0260 | 0.0272 | 0.0128 |
| CResT (1 stage) | **0.7322** | **0.4061** | **0.6420** | **0.3935** |
| No Connection | 0.1466 | 0.0556 | 0.0393 | 0.0245 |
| All Connections | **0.7312** | **0.4073** | **0.6254** | **0.3930** |
| No Mask | 0.7118 | 0.4003 | 0.5997 | 0.3800 |
| Random Mask | **0.7312** | **0.4073** | **0.6254** | **0.3930** |

passes Spatial Attention with 32.72% higher success rate and 0.2678 higher SPL in all episodes, and 34.56% higher success rate and 0.2628 higher SPL in $L \geq 5$ episodes.

**Comparing to SAVN Wortsman et al. (2019) baseline.** SAVN uses a self-adaptive visual navigation method (SAVN) to adapt to new environments easily in testing. However, SAVN obtain limited performance because the visual coding network is only a trained ResNet18. With the mighty visual coding network, CResT significantly outperforms SAVN with 29.72% higher success rate and 0.2451 higher SPL in all episodes, and 35.01% higher success rate and 0.2569 higher SPL in $L \geq 5$ episodes on AI2-THOR dataset.

**Comparing to ORG and ORG+TPN Du et al. (2020a) baseline.** ORG uses the object representation graph (ORG) to encode detection features and employs ResNet18 He et al. (2016) to extract the global feature, and the features are fused to the final visual representation. This visual encoding network enables ORG to greatly exceed SAVN and Spatial Attention, and TPN helps the agent escape from deadlock, which can also obtain small improvement. However, its ability of visual information processing is still not as good as VTNet and CResT. CResT outperforms ORG with 6.38% higher success rate and 0.0629 higher SPL in all episodes, and 7.2% higher success rate and 0.067 higher SPL in $L \geq 5$ episodes. CResT also surpasses ORG+TPN for 6.14% higher success rate and 0.0222 higher SPL in all episodes, and 7.59% higher success rate and 0.0296 higher SPL in $L \geq 5$ episodes.

**Comparing to VTNet Du et al. (2020b) baseline.** VTNet uses the Visual Transformer to encode both the local feature and global feature and exceeds all previous competing methods. However, VTNet requires two-stage training due to its sophisticated visual encoding network with the standard Transformer, the result of which is inferior to that of one-stage training. CResT enables the network to be trained once in the deep network because of the Residual Transformer proposed in this paper. In the experiment of the same condition and hyperparameters, the CResT exceeds VTNet with 3.46% higher success rate in all episodes, and 5.89% higher success rate and 0.0043 higher SPL in $L \geq 5$ episodes, which makes CResT surpass VTNet and other competing methods and achieve state-of-the-art performance on AI2-THOR dataset.

## 4.5 ABLATION EXPERIMENTS

The ablation study is also performed to manifest the impact of different components in CResT, as shown in Table 2. The contribution of the Residual Transformer is proved in the ablation experiments of Transformer layers, Residual connections, and Training in one stage, and the effect of Random Mask is stated in the ablation experiments of Random Mask. In the ablation experiments, the default mask threshold and layer number of the Transformer are 0.1 and 1, respectively.

**Transformer layers.** The performance of CResT improves slightly when the number of Transformer layers increases, and the gradient vanishing problem does not appear when the network is deepened. This experiment shows the Residual Transformer effectively enhances the backpropagation and solves the gradient vanishing problem in the Transformer.

**Training in one stage.** Although VTNet obtains relatively satisfying performance in the two-stage training, its navigation result in the one-stage training is quite poor, which is 8.45% success rate and 0.0260 SPL in all episodes, and 2.72% success rate and 0.0128 SPL in $L \geq 5$ episodes, which seems almost untrained compared to the result of CResT trained in the same condition and hyperparameters.

**Residual connections in the Residual Transformer.** The No Connection method is the CResT with the standard Transformer, the performance of which is quite unsatisfying because of the gradient vanishing problem. The All Connections method is the CResT with the Residual Transformer which achieves excellent performance. These experiments prove the efficiency of the Residual Transformer and its residual connections.

**Random Mask.** The CResT with the Random Mask (Mask threshold = 0.1) obtains better performance compared to it with No Mask (Mask threshold = 0), which proves the Random Mask improves the Local Feature and the Global Feature in the way of data augmentation and therefore achieves better performance in navigation.

## 5 CONCLUSION

In this paper, we propose the Cross-Query Residual Transformer (CResT) for visual information processing. In the overall method, the Global Feature and Local Feature query each other and are fused for better visual feature extraction. The Residual Transformer is proposed to solve the gradient vanishing problem and enable the network to be trained in one stage. The Random Mask is proposed to improve visual features in the way of data augmentation, which reduces overfitting and enhances the generalization performance of the network. The experimental results show that CResT surpasses the competing methods in the object goal navigation and achieves the state-of-the-art method, and the ablation experiments prove the Residual Transformer and the Random Mask efficiently improve the navigation results.

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
