# OpenReview forum: "CResT: Cross-Query Residual Transformer for Object Goal Navigation"
_ICLR.cc/2024/Conference — Submitted to ICLR 2024_

### Official Review · Reviewer_divX · 2023-10-29

**Soundness:** 1 poor
**Presentation:** 1 poor
**Contribution:** 1 poor
**Rating:** 3
**Confidence:** 5

**Summary:**

This paper proposes an end-to-end Transformer-based architecture for object goal navigation. The main contribution is the inclusion of additional residual connections in the transformer, which helps in training a deeper architecture. The method is evaluated in the AI2-THOR synthetic environment.

**Strengths:**

The proposed residual connections might actually be helpful, but further investigation is required.

**Weaknesses:**

Limited novelty, writing needs improvement, experiments are insufficient. See comments underneath for details.

**Questions:**

The intro reads more as a related work section (particularly paragraph 3). It is not properly explained what is the limitation of the prior work that this paper tries to address. There is no reason to explain why the state representation is important for end-to-end methods since that is already kinda obvious. Rather give examples of what existing state representations are not able to encode well, in order to build your narrative towards the paper's contribution. Similarly, the contributions of the paper are not clear in the intro. For example, in what ways is the visual information from the proposed CResT better?
The writing needs substantial improvements. Statements such as "we propose this because it is better" (without further explanation) are uninformative and should be avoided. Also there are minor language issues throughout the paper. One example: first sentence of the intro "navigating" -> "navigation".

The paper presents the Residual Transformer as one of its contributions which is confusing, especially for an object-goal navigation paper. The authors propose three residual connections to the vanilla transformer architecture. But various designs on residual connections for transformers have already been explored in the literature which are not cited in the paper:
[A] He et al, RealFormer: Transformer Likes Residual Attention, ACL 2021
[B] Xie et al, ResiDual: Transformer with Dual Residual Connections, arXiv 2023
If the authors are proposing a new design of their own in order to address the gradient vanishing problem (which they keep mentioning) then they should follow the methodologies of [A,B] on how to present and justify their proposal. In the context of object goal navigation, the proposed design is a straightforward extension of transformers without proper justification. This is exacerbated by the trivial description of the overall architecture. It seems to me that the inspiration for this work is the VTNet paper (which is cited and compared to in the paper). However, in VTNet there is a clear description of the thought process behind the architecture and visual demonstration of how the resulting representation is useful for object-goal navigation.

Random mask for data augmentation is pretty common and should not be claimed as novelty.

There are some discrepancies in the experimental results. First, the performance reported in the VTNet paper is higher than what is shown here. If the authors followed a different experimental setup than the baselines then it should be clearly noted in the paper. Second, in the ablation experiment of No connections / All connections, the vanilla transformer is shown to fail (14.6% SR), which should encourage more investigation on the experiment itself. Vanilla transformers are already very powerful in extracting visual representations and have been used with relative success in obj-nav including the VTNet paper so I used expect at least decent performance.

Finally, if the main motivation of the paper is to enhance visual features, then perhaps a more realistic environment (e.g., Habitat) is more suitable to demonstrate the work, rather than a fully synthetic environment.

---

### Official Review · Reviewer_oDQU · 2023-10-31

**Soundness:** 2 fair
**Presentation:** 2 fair
**Contribution:** 1 poor
**Rating:** 3
**Confidence:** 5

**Summary:**

This submission introduces the Cross-Query Residual Transformer for Object Goal Navigation. The proposed method mainly consists of Residual Transformer and Random Mask. The proposed method is evaluated on the AI2-THOR dataset. The submission claims to achieve state-of-the-art performance. However, the methods in comparison seem quite outdated.

**Strengths:**

- This submission clearly presents the ObjectNav task.
- This submission attempts to utilize information at different scales (global and local).

**Weaknesses:**

- The technical contributions are somewhat weak; several claimed innovations in the paper, such as the Transformer and Residual connections, are well-established methods that have been previously introduced and proven effective.
- The work appears more akin to a general visual extractor and does not seem particularly tailored to the ObjectNav task.
- The submission lacks comparison with many recent works conducted on the AI2-THOR simulator; the following is a subset.
	[1] Object-Goal Visual Navigation via Effective Exploration of Relations among Historical Navigation States. CVPR 2023
	[2] Layout-Based Causal Inference for Object Navigation. CVPR 2023
	[3] Search for or Navigate to? Dual Adaptive Thinking for Object Navigation. ICCV 2023
- The idea lacks novelty. The method overall seems to merely merge the global and object features provided by VTNet in a slightly modified form, and the performance improvement compared to VTNet is not significant.
- The experimental section is quite insufficient, lacking adequate demonstration of the motivation, making it difficult to substantiate the method's effectiveness and necessity.

**Questions:**

- The submission declares that "The Residual Transformer is proposed to solve the gradient vanishing problem," It's unclear how this gradient vanishing problem' manifests uniquely in the ObjectNav task. Are there experimental results or analyses that demonstrate this phenomenon in the ObjectNav task? Evidence of gradient vanishing specifically impacting the performance or training stability in ObjectNav would be crucial to justify the adoption of the Residual Transformer as a solution.
- In the submission, the motivation behind adopting a cross-query approach is not clear. Why is it implemented only in the decoder and not in the encoder? The submission should provide evidence or theoretical justification for these design choices.
- The one of contribution of the paper, "Random Mask", seems to be an operation similar to dropout?
- Are there specific examples or evidence in the context of "Overfitting means wonderful training results yet unsatisfying testing results, and it becomes obvious when a relatively larger network trains on a relatively limited dataset."? For instance, are there experimental results or visualizations demonstrating this phenomenon in the ObjectNav task?

---

### Official Review · Reviewer_9zLV · 2023-10-31

**Soundness:** 2 fair
**Presentation:** 2 fair
**Contribution:** 2 fair
**Rating:** 3
**Confidence:** 4

**Summary:**

The document introduces the Cross-Query Residual Transformer (CResT) method for object goal navigation. CResT enhances navigation by improving visual feature extraction from RGB images. The architecture incorporates an overall network with Global and Local Features, a Residual Transformer for deeper training, and a Random Mask for data augmentation. The method surpasses other models on the AI2-THOR dataset.

**Strengths:**

1. The authors introduce the Cross-Query Residual Transformer (CResT), which excels in the extraction of intricate visual features from RGB camera images. As a result, CResT provides a more holistic and informative visual representation. This innovation allows CResT to achieve state-of-the-art performance on the AI2-THOR dataset in the realm of object goal navigation.
2. The proposed CResT can be trained in one stage instead of the two-stage training in the previous method. This makes the training process more concise and efficient.
3. The authors also adopt a Random Mask module to augment the Global Feature and Local Feature. This data augmentation technique augments both the Global and Local Features of the network, thereby improving its generalization capabilities and effectively mitigating the risk of overfitting.

**Weaknesses:**

1. Lack of in-depth explanation of the contributions: The authors do not provide sufficient insights into the introduction of residual connections in the Transformer. It would be beneficial to have a more detailed explanation of how these connections address the gradient vanishing problem and improve the backpropagation in the network.
2. Limited novelty in the proposed random mask module: The proposed random mask module is a commonly used data augmentation technique in deep learning. While it is applied in the context of visual navigation, the paper does not provide enough justification or explanation for its effectiveness specifically for this task. Further analysis or comparison with alternative data augmentation methods would strengthen the novelty of this module.

**Questions:**

The primary question concerning the novelty of the proposed methods has been articulated in the weaknesses section. If the authors successfully address the concerns specified in the weaknesses section, I would reconsider my recommendation.

---

### Official Review · Reviewer_4dQB · 2023-11-04

**Soundness:** 3 good
**Presentation:** 3 good
**Contribution:** 2 fair
**Rating:** 3
**Confidence:** 5

**Summary:**

This article aims to improve the end-to-end object navigation system. The author focuses on improving the visual state representation of the navigator. Specifically, the author introduces additional shortcut connections to the transformer architecture to alleviate the gradient vanishing problem and applies random masking to the visual features as data augmentation. The proposed method is validated on the AI2-THOR dataset.

**Strengths:**

The research problem is interesting and holds broad potential value due to the significance of visual representations in embodied tasks.

**Weaknesses:**

1. The technique contribution seems limited and appears to be an incremental improvement to VTNet by introducing more shortcut connections and random masking. I am not convinced by the author's statement regarding the gradient vanishing problem in the original transformer. The reason is as follows:
a) The author's argument is solely based on the improved performance of object navigation after adding more shortcuts. It lacks supporting theoretical, visualization, or metric analysis.
b) In Table 2, the performance of a 1-layer transformer is already good (0.7312 SR), and the improvement with a 6-layer transformer is limited (0.7322 SR). How does this example demonstrate the effectiveness of adding more shortcuts to address the gradient vanishing problem?
2. The experiments are insufficient as they are only validated on the AI2-THOR dataset. Why weren't other object navigation datasets, such as Habitat [1] or ProcTHOR [2], used for validation?
3. The comparative methods are insufficient, and I hardly see any recent object navigation works from 2022 and 2023.

[1] Habitat-web: Learning embodied object-search strategies from human demonstrations at scale, CVPR’22
[2] ProcTHOR: Large-Scale Embodied AI Using Procedural Generation, NeurIPS’22

**Questions:**

What are the advantages of studying end-to-end object navigation when recent articles [3] have shown that it fails when transferred to the real world, compared to the counterpart modular approaches?

[3] Navigating to objects in the real world, Science Robotics’23

---

### Meta-Review · Area_Chair_8CxN · 2023-12-03

**Metareview:**

The paper has received four reject ratings. The reviewers have concerns regarding the novelty of the paper, the writing, insufficient comparisons with prior work, and weak technical contributions. The reviewers have consensus both on the rating and the concerns. The authors did not submit a rebuttal to address the concerns. Therefore, the AC follows the recommendation of the reviewers and recommends rejection.

**Justification For Why Not Higher Score:**

The authors did not address the reviewers’ concerns.

**Justification For Why Not Lower Score:**

N/A

---

### Decision · Program_Chairs · 2024-01-16

Reject